# Study on the Evaluation of Emergency Management Capacity of Resilient Communities by the AHP-TOPSIS Method

**DOI:** 10.3390/ijerph192316201

**Published:** 2022-12-03

**Authors:** Kai Wang, Zhe Wang, Jun Deng, Yuanyuan Feng, Quanfang Li

**Affiliations:** 1College of Safety Science and Engineering, Xi’an University of Science and Technology, Xi’an 710054, China; 2Xi’an Key Laboratory of Urban Public Safety and Fire Rescue, Xi’an 710054, China

**Keywords:** AHP-TOPSIS, resilient communities, emergency management, indicator system, evaluation model

## Abstract

Community emergency management is directly related to the safety of people’s lives and properties and is concerned with economic development and social stability. This paper established an evaluation model of community emergency management capacity from the perspective of resilience based on hierarchical analysis (AHP) and distance between superior and inferior solutions (TOPSIS). In terms of infrastructure resilience, community organizational resilience, risk, and hidden danger management, emergency material security, emergency force construction, emergency literacy, and the evaluation index system of resilient community emergency management capacity were improved. By the AHP method, the weights of all indexes were determined scientifically. Combined with the TOPSIS method, the fit of the evaluated object which between the positive and negative ideal solution was calculated to determine the optimal evaluation among multiple experts. According to the validated assessment, the low-scoring indicators were analyzed to make practical suggestions for improvement. The results provide new theoretical methods and technical support for the assessment of community emergency response capacity, which also provides reference for the assessment of emergency response capacity in other fields.

## 1. Introduction

In recent years, with the rapid growth of the global population, uncertain disasters and emergencies, such as those meteorological, geological, or epidemic in origin, have occurred in many places around the world and show a trend of normalization [1,2,3,4,5]. They cause psychological trauma and property damage to human beings; they also seriously lead to an imbalance of social order and irreparable catastrophic consequences. In the most disaster-prone area communities, due to the characteristics of a concentrated population, concentrated buildings, full production, and full information, an irreversible situation will be formed once a disaster is encountered [6,7,8,9]. Resilient community construction provides a new idea to improve the emergency management capacity of communities so that communities can prepare for crises and avoid large shocks to communities [10,11]. When the problem is over, the community can quickly return to average production and living levels, thus ensuring community stability.

Currently, scholars are increasingly focusing on the resilience perspective in community emergency management systems. Resilience is a core element of disaster risk reduction [12,13,14]. Generally, a resilient system is defined as a flexible structure that can quickly adapt the system to a changing environment and mobilize all forces broadly to respond to emergencies [15,16,17,18].

There has been a great deal of research into the factors influencing community resilience. Ainuddin et al. [19] emphasized that the human element should be given prominence and priority, and that only when human initiative is mobilized, community emergency management capacity resilience can be fundamentally improved, and that residents’ reorganizational capacity and collective behavior are important factors that affect post-disaster community resilience enhancement. Anders Oskarsson et al. [20] proposes nine areas of interaction, coordination, decision-making, relationships, awareness, resilience, preparedness, system performance, and information infrastructure to improve the emergency management capacity of resilient communities. Alonge et al. [21] and Igalla et al. [22] suggest that strong leadership, strong links up and down the hierarchy, and effective communication are strong safeguards for resilient communities when responding to public crisis events.

However, in terms of emergency management capacity assessment methods, Pfefferbaum et al. [23] conducted a resilience survey study of five communities’ emergency management capacity by applying the community advanced resilience model (CART). The results showed that the model could be directly applied to the real world, and the model was used to assess community resilience and guide community resilience-building based on the information collected by the community. Wang et al. [24] established a community emergency management capacity evaluation index system based on the emergency management cycle theory, and assessed community resilience based on entropy weight and a multi-layer fuzzy integrated evaluation model. Zhang et al. [25] explored a BP-neural network-based emergency logistics capacity evaluation model from the perspective of COVID-19. In terms of research on multi-criteria decision-making methods, AHP has been widely used in combination with other evaluation methods, such as hierarchical analysis, fuzzy integrated evaluation methods [26], Entropy-AHP, GIS [27], AHP 2-tuple fuzzy linguistic [28], R-AHP [15], FAHP-FDEMATEL-TOPSIS [29], and so on.

In the last decade of research, many applications of AHP-TOPSIS have been published in international scientific articles, and the combination of AHP and TOPSIS allows for solid factor weight determination and decision-making. Well-proven in many areas, such as assessing the quality factors of banking e-services [30], addressing the selection of bank branch locations [31], and application to the assessment of ergonomic risk factors [32], addressing the selection of suppliers in the electronics industry, textile industry, manufacturing industry, and the pharmaceutical industry [33,34,35,36,37,38]. For some complex assessment objects, the AHP-TOPSIS method has also been well demonstrated e.g., Yoon et al. [39] assessed the nuclear fuel cycle based on five main assessment factors. Bakioglu and Atahan [40] developed an effective risk assessment process for autonomous vehicles.

Based on the previous study, four significant advantages of the AHP-TOPSIS method can be summarized: (1) good logic, (2) reflecting both positive and negative desirable options, (3) simple calculation steps and process, (4) TOPSIS provides efficiency in ranking compared to other methods [36,41]. This study aims to apply the AHP-TOPSIS method innovatively to evaluate the emergency management capacity of resilient communities. The method provides different factor weights for the assessment object and in-depth analysis based on the best and worst ideal solution scenarios [42,43].

## 2. Community Emergency Management Capacity Evaluation Model

This section describes the evaluation process of the proposed AHP-TOPSIS method and the methodology used to analyze the results of the case study. Firstly, Figure 1 shows the main steps of the evaluation process. The key features of each step are then described in detail.

### 2.1. AHP Method and Its Principle

The analytic hierarchy process (AHP) is an analytical method for multi-attribute decision problems. The method was introduced in the early 1970s. According to the nature of the problem and the overall objective to be achieved, the problem is decomposed into different constituent factors, and other level sets are formed according to the interrelated influence and affiliation of the elements, including a multi-level analysis structure model [44]. The specific calculation steps are as follows.

(1)Construction of two-two judgment matrix

In the hierarchical structure of the index system, the importance of the secondary indicators of the same affiliation under each primary hand is judged separately and assigned according to a scale of 1–9. The judgment scales are defined as shown in Table 1.

(2)The root method is used to calculate *λ_max_* and *w_i_*

As in Equation (1), the elements of the judgment matrix are multiplied by rows, and then the resulting products are rooted n times, respectively.
(1)$$w_{i}^{*}=\sqrt[{n}]{\left(w_{1j}^{*}\cdot w_{2j}^{*}\cdot \cdot \cdot \cdot \cdot \cdot \cdot w_{nj}^{*}\right)}$$
where $w_{i}^{*}$ is the geometric mean of the elements of each row in the judgment matrix; $w_{nj}^{*}$ is the assigned value of the elements in the judgment matrix; *n* is the order of the judgment matrix.

(3)Normalize the square root vector

(2)$$w_{i}=\frac{w_{i}^{*}}{w_{1}^{*}\cdot w_{2}^{*}\cdot \cdot \cdot \cdot \cdot \cdot w_{n}^{*}}$$
where $w_{i}$ is the geometric mean of the elements of each row after normalization.

(4)Calculate eigenvalues and perform consistency tests

(3)$$\lambda =\frac{1}{n}\left(\frac{Aw_{1}}{w_{1}}+\frac{Aw_{2}}{w_{2}}+\cdot \cdot \cdot \cdot \cdot \cdot +\frac{Aw_{n}}{w_{n}}\right)$$
where λ is the maximum eigenvalue; A is the constructed judgement matrix.

Calculation of consistency metrics
(4)$$CI=\frac{\lambda -n}{n-1}$$

The smaller the calculated *CI* value, the better the consistency of the constructed judgment matrix; conversely, the larger the *CI* value, the worse the character.

Calculate the consistency ratio.
(5)$$C.R.=\frac{CI}{RI}<0.1$$
where *RI* is the random consistency index as shown in Table 2.

### 2.2. Evaluation Model by TOPSIS Method

The ranking method of approximating ideal values (TOPSIS) is a multi-attribute decision analysis method. Its basic principle is to rank the evaluation object by detecting the distance between the object and the optimal solution and the worst solution, if the object is close to the optimal solution and far from the worst solution at the same time, it is the best; if it is the opposite, it is the worst [41,45,46,47,48,49]. By calculating the level value, the optimal solution, and the worst solution of community emergency management capability, calculating the distance between the level value with the optimal solution and the worst solution respectively, and finally comparing and ranking the evaluation results, the most likely evaluation result of community emergency management capability is determined.

(1)Calculate the weighting matrix

The decision matrix constructed from the expert scoring data was multiplied by the weights of each indicator to obtain the weighting matrix *R* = (*r_ij_*)*_m×n_*.
(6)$$r_{ij}=\omega_{j}\cdot b_{ij}\;\left(i=1,\;2\ldots \ldots ,\;m;\;j=1,\;2\ldots \ldots ,n\right)\;$$
where $\omega_{j}$ is the weight of the *j* indicator; $b_{ij}$ is the decision matrix constructed from the expert scoring data.

(2)Determine the positive and negative ideal solution.
(7)$$S_{j}^{+}=\underset{1\leq i\leq n}{\max}\left\{r_{ij}\right\},j=1,2,3,\ldots ,m$$
(8)$$S_{j}^{-}=\underset{1\leq i\leq n}{\min}\left\{r_{ij}\right\},j=1,2,3,\ldots ,m$$
where $S_{j}^{+}$ is a positive ideal solution; $S_{j}^{-}$ is a negative ideal solution.(3)Calculate the separation of a certain set of expert scoring data with the positive ideal solution and the negative ideal solution.
(9)$$d_{i}^{+}=\sqrt{\sum_{j=1}^{n}{\left(r_{ij}-S_{j}^{+}\right)}^{2}}$$
(10)$$d_{i}^{-}=\sqrt{\sum_{j=1}^{n}{\left(r_{ij}-S_{j}^{-}\right)}^{2}}$$
where $d_{i}^{+}$ is the Euclidean distance between the evaluation sample and the positive ideal solution; $d_{i}^{-}$ is the Euclidean distance between the evaluation sample and the negative ideal solution.(4)Calculate the relative proximity of each group of experts’ scores and the positive ideal solution.
(11)$$\sigma_{i}=\frac{d_{i}^{-}}{d_{i}^{+}+d_{i}^{-}}\left(i=1,\;2,\;\ldots \ldots ,\;m\right)\;$$

The relative proximity of the scores of each group is determined according to the value, and the higher value indicates that the *i* group of experts’ scores is closer to the positive ideal solution, meaning the better the scoring result. Therefore, the optimal scoring result of the community can be determined by determining the largest group of experts’ scoring data.

## 3. Case Study—A Community in Xi’an City as an Example

### 3.1. Indicator System Construction

For community resilience in the context of public emergencies, it is necessary to consider the importance of and factors influencing the response in the pre-disaster event (including community infrastructure, community safety culture, risk identification, etc.), mid-disaster (including rescue workers, service volunteers, material deployment, etc. in the disaster management process), and post-disaster (including post-disaster community reconstruction, degree of recovery, etc.). In addition, emergency assessment is crucial to disaster relief. Therefore, in the process of establishing the indicator system, we used the “community survey—literature data collection—expert assessment” method to develop comprehensive evaluation indicators. The process of establishing the evaluation index system is shown in Figure 2.

(1)Community visits and surveys

Most evaluations of resilient community emergency management capacity rely on quantitative secondary indicator data, but the inclusion of perceptions can add more context-specific factors [50,51]. During the community interviews, we invited relevant community managers, long-time community residents, and merchant practitioners who had a comprehensive understanding of the community and its surrounding environment to take part. The core content of the interviews included both external environmental factors (community service capacity, community infrastructure development, community safety culture development, community management capacity, etc.) and the emergency response capacity that one possesses (education level, safety awareness, psychological condition, etc.).

(2)Documentary data collection

The strategy of searching the literature on emergency management capacity evaluation indicator systems for resilient communities consisted of a comprehensive search by accessing the Wanfang Data Knowledge Service Platform (Wanfang Data), the China Knowledge Resources Database (CNKI), CSI databases, and emergency management literature databases, with the search period restricted to 2012 to 2022. During this period, the Endnote web literature management software was used to categorize and summarize the literature found and to identify frequently occurring key indicators. In addition, by combining the results of the visitor survey and the document “National Comprehensive Disaster Reduction Demonstration Community Standards” [52] issued by the National Disaster Reduction Committee, six aspects, such as infrastructure, organization and management, risk and hazard, emergency materials, emergency response force, and emergency literacy, were initially selected as the general framework of the indicator system, followed by a refined division of each indicator to establish the corresponding secondary indicators.

(3)Expert assessment

Experts with a high level of knowledge of the relevant neighborhoods as well as academics with extensive experience were invited to assess the evaluation indicator system [53,54]. The experts’ assessment includes the overlap of indicators, the importance of indicators, and the attributes of indicators [29,51]. Through two rounds of assessment by the expert group, a community emergency management capacity evaluation index system with 6 primary indicators and 25 secondary indicators under the resilience perspective was established, as shown in Figure 3.

### 3.2. Determination of Index Weight

The judgment matrix was determined by inviting experts to judge the importance of the second-level indicators of the same affiliation under each first-level hand separately. Table 3 shows the judgment matrix constructed for the first-level hands.

The result of the calculation of the largest eigenvalue of the judgment matrix is λ = 6.621;

The result of the consistency index is $CI=\frac{6.521-6}{6-1}=0.1042$;

Table 2 shows that the average random consistency index is *RI* = 1.26. The random consistency rate is: $C.R.=\frac{CI}{RI}=\frac{0.1042}{1.26}=0.0827<0.1$.

Therefore, the judgment matrix constructed according to the first-level index passes the test of consistency, and the result of the weight value calculated by AHP is reasonable.

The weights of the second-level indicators were calculated according to the calculation method of the weights of the first-level indicators. Then the indicators at each level were dimensionless processed to obtain the total weights. The finalized relative weights of community emergency management capacity evaluation indicators under the resilience perspective are shown in Figure 4.

### 3.3. TOPSIS Evaluation Model

Selecting stakeholders with a high level of knowledge of the evaluation system to assess indicators produces good results [53,55]. Secondly, at the local level, the opinions of practitioners and policy advisors with extensive experience in emergency management are key in assessing community emergency management capacity [51,54]. Based on this, a total of nine experts in the relevant fields were invited to assign scores to each secondary indicator.

Indicators are scored in a range of 1 to 10, with higher scores indicating better implementation of the indicator and lower scores indicating that the indicator needs to be further strengthened.

The decision matrix *P* was obtained by standardizing and normalizing the scoring data as follows.
$$P=\left[\begin{array}{ccccccccc} 0.0136 & 0.0091 & 0.0106 & 0.0125 & 0.0200 & 0.0077 & 0.0096 & 0.0113 & 0.0102\\ 0.0097 & 0.0091 & 0.0106 & 0.0143 & 0.0150 & 0.0087 & 0.0086 & 0.0127 & 0.0127\\ 0.0155 & 0.0104 & 0.0106 & 0.0143 & 0.0125 & 0.0077 & 0.0077 & 0.0127 & 0.0127\\ 0.0117 & 0.0078 & 0.0094 & 0.0125 & 0.0150 & 0.0087 & 0.0096 & 0.0099 & 0.0102\\ 0.0136 & 0.0104 & 0.0094 & 0.0107 & 0.0150 & 0.0096 & 0.0096 & 0.0071 & 0.0102\\ 0.0155 & 0.0117 & 0.0094 & 0.0143 & 0.0125 & 0.0096 & 0.0077 & 0.0014 & 0.0089\\ 0.0117 & 0.0065 & 0.0082 & 0.0125 & 0.0100 & 0.0077 & 0.0086 & 0.0085 & 0.0089\\ 0.0155 & 0.0091 & 0.0106 & 0.0107 & 0.0125 & 0.0096 & 0.0096 & 0.0127 & 0.0114\\ 0.0097 & 0.0104 & 0.0082 & 0.0143 & 0.0150 & 0.0077 & 0.0077 & 0.0127 & 0.0127\\ 0.0155 & 0.0078 & 0.0094 & 0.0089 & 0.0150 & 0.0087 & 0.0086 & 0.0113 & 0.0102\\ 0.0136 & 0.0117 & 0.0106 & 0.0107 & 0.0100 & 0.0077 & 0.0067 & 0.0099 & 0.0089\\ 0.0136 & 0.0104 & 0.0082 & 0.0107 & 0.0125 & 0.0096 & 0.0096 & 0.0085 & 0.0089\\ 0.0117 & 0.0117 & 0.0094 & 0.0071 & 0.0125 & 0.0096 & 0.0096 & 0.0085 & 0.0102\\ 0.0097 & 0.0104 & 0.0106 & 0.0107 & 0.0150 & 0.0096 & 0.0077 & 0.0141 & 0.0127\\ 0.0097 & 0.0091 & 0.0106 & 0.0125 & 0.0175 & 0.0096 & 0.0096 & 0.0141 & 0.0127\\ 0.0097 & 0.0104 & 0.0094 & 0.0089 & 0.0150 & 0.0077 & 0.0077 & 0.0099 & 0.0063\\ 0.0117 & 0.0117 & 0.0082 & 0.0125 & 0.0175 & 0.0087 & 0.0086 & 0.0141 & 0.0102\\ 0.0136 & 0.0104 & 0.0094 & 0.0089 & 0.0125 & 0.0087 & 0.0086 & 0.0099 & 0.0089\\ 0.0117 & 0.0091 & 0.0106 & 0.0143 & 0.0150 & 0.0087 & 0.0086 & 0.0099 & 0.0063\\ 0.0136 & 0.0117 & 0.0082 & 0.0125 & 0.0175 & 0.0087 & 0.0096 & 0.0099 & 0.0089\\ 0.0155 & 0.0117 & 0.0106 & 0.0107 & 0.0125 & 0.0096 & 0.0096 & 0.0085 & 0.0063\\ 0.0117 & 0.0104 & 0.0082 & 0.0125 & 0.0100 & 0.0077 & 0.0077 & 0.0085 & 0.0102\\ 0.0097 & 0.0091 & 0.0094 & 0.0107 & 0.0150 & 0.0087 & 0.0096 & 0.0085 & 0.0089\\ 0.0097 & 0.0104 & 0.0106 & 0.0125 & 0.0125 & 0.0096 & 0.0096 & 0.0113 & 0.0102\\ 0.0097 & 0.0117 & 0.0106 & 0.0143 & 0.0100 & 0.0087 & 0.0077 & 0.0113 & 0.0102 \end{array}\right]$$

The weighting matrix *R* is calculated according to Equation (6).
$$R=\left[\begin{array}{ccccccccc} 0.00136 & 0.00091 & 0.00106 & 0.00125 & 0.00200 & 0.00077 & 0.00096 & 0.00113 & 0.00102\\ 0.00046 & 0.00043 & 0.00050 & 0.00068 & 0.00071 & 0.00041 & 0.00041 & 0.00060 & 0.00060\\ 0.00089 & 0.00060 & 0.00061 & 0.00082 & 0.00072 & 0.00044 & 0.00044 & 0.00073 & 0.00073\\ 0.00029 & 0.00020 & 0.00024 & 0.00031 & 0.00038 & 0.00022 & 0.00024 & 0.00025 & 0.00025\\ 0.00027 & 0.00021 & 0.00019 & 0.00021 & 0.00030 & 0.00019 & 0.00019 & 0.00014 & 0.00020\\ 0.00098 & 0.00074 & 0.00059 & 0.00090 & 0.00079 & 0.00061 & 0.00048 & 0.00009 & 0.00056\\ 0.00051 & 0.00028 & 0.00036 & 0.00054 & 0.00044 & 0.00034 & 0.00038 & 0.00037 & 0.00039\\ 0.00028 & 0.00016 & 0.00019 & 0.00019 & 0.00023 & 0.00017 & 0.00017 & 0.00023 & 0.00021\\ 0.00025 & 0.00027 & 0.00021 & 0.00036 & 0.00038 & 0.00020 & 0.00020 & 0.00032 & 0.00032\\ 0.00228 & 0.00115 & 0.00139 & 0.00131 & 0.00221 & 0.00127 & 0.00127 & 0.00166 & 0.00149\\ 0.00057 & 0.00049 & 0.00045 & 0.00045 & 0.00042 & 0.00032 & 0.00028 & 0.00042 & 0.00037\\ 0.00128 & 0.00099 & 0.00078 & 0.00101 & 0.00118 & 0.00091 & 0.00091 & 0.00080 & 0.00084\\ 0.00077 & 0.00078 & 0.00063 & 0.00048 & 0.00083 & 0.00064 & 0.00064 & 0.00056 & 0.00068\\ 0.00052 & 0.00056 & 0.00057 & 0.00057 & 0.00080 & 0.00051 & 0.00041 & 0.00075 & 0.00068\\ 0.00037 & 0.00034 & 0.00040 & 0.00047 & 0.00066 & 0.00036 & 0.00036 & 0.00053 & 0.00048\\ 0.00013 & 0.00014 & 0.00012 & 0.00012 & 0.00020 & 0.00010 & 0.00010 & 0.00013 & 0.00008\\ 0.00030 & 0.00031 & 0.00021 & 0.00033 & 0.00046 & 0.00023 & 0.00022 & 0.00037 & 0.00026\\ 0.00024 & 0.00019 & 0.00017 & 0.00016 & 0.00023 & 0.00016 & 0.00016 & 0.00018 & 0.00016\\ 0.00015 & 0.00011 & 0.00013 & 0.00018 & 0.00019 & 0.00011 & 0.00011 & 0.00012 & 0.00008\\ 0.00011 & 0.00009 & 0.00007 & 0.00010 & 0.00014 & 0.00007 & 0.00008 & 0.00008 & 0.00007\\ 0.00011 & 0.00008 & 0.00007 & 0.00008 & 0.00009 & 0.00007 & 0.00007 & 0.00006 & 0.00004\\ 0.00005 & 0.00005 & 0.00004 & 0.00006 & 0.00005 & 0.00003 & 0.00003 & 0.00004 & 0.00005\\ 0.00011 & 0.00010 & 0.00010 & 0.00012 & 0.00017 & 0.00010 & 0.00011 & 0.00009 & 0.00010\\ 0.00020 & 0.00022 & 0.00022 & 0.00026 & 0.00026 & 0.00020 & 0.00020 & 0.00024 & 0.00021\\ 0.00037 & 0.00044 & 0.00040 & 0.00054 & 0.00038 & 0.00033 & 0.00029 & 0.00043 & 0.00038 \end{array}\right]$$

From Equations (7) and (8), the values of positive ideal solution $S_{j}^{+}$ and negative ideal solution $S_{j}^{-}$ are calculated.
$$\begin{array}{cc} S_{j}^{+}= & [0.00136\;0.00071\;0.00089\;0.00038\;0.00030\;0.00098\;0.00054\;0.00028\;0.00038\;0.00228\;0.00057\;0.00128\;0.00078\;0.00080\\ & 0.00066\;0.00020\;0.00046\;0.00024\;0.00019\;0.00014\;0.00011\;0.00006\;0.00017\;0.00026\;0.00054{{{]}_{1}}_{\times }}_{25}\\ S_{j}^{-}= & [0.00077\;0.00041\;0.00044\;0.00020\;0.00014\;0.00048\;0.00034\;0.00016\;0.00020\;0.00115\;0.00028\;0.00078\;0.00048\;\\ & 0.00041\;0.00034\;0.00010\;0.00021\;0.00016\;0.00008\;0.00007\;0.00004\;0.00003\;0.00009\;0.00020\;0.00029{{{]}_{1}}_{\times }}_{25} \end{array}$$

The closeness of the nine sets of scoring data to the ideal solution is calculated according to Equations (9)–(11).
$$\sigma_{i}={\left[0.7398\;0.2921\;0.2877\;0.4645\;0.7195\;0.1638\;0.1681\;0.4228\;0.3699\right]}_{1\times 9}$$

The larger the value is, the closer it is to the positive ideal solution. It can be seen that the maximum value among the nine sets of data is 0.7398, which corresponds to the score of the first expert, that is, the first expert scores are the most reasonable. By analyzing the score of this expert, the most reasonable evaluation result of the community’s emergency management capability can be obtained.

### 3.4. Evaluation Results and Discuss

In order to obtain more intuitive, realistic, and comprehensive model evaluation results, two weighting techniques of AHP and TOPSIS were utilized. The two techniques have been rapidly developed in recent years [15,24,26,27,28,29]. In this paper, the contribution of the study is extended by combining the use of AHP and TOPSIS to assess subjectively and objectively the weights of indicators considered relevant to improving community emergency management capacity. In addition, the addition of practitioner indicators (such as disaster information officer and clearance group in this paper’s research, which had not been considered in previous studies) increases the reliability of the findings and reflects the important role of practitioners in building community emergency management capacity.

The scoring of each index by the first expert was plotted as a bar chart, as shown in Figure 5.

It can be visually found from the graph that there are obvious shortcomings in the construction of emergency response capacity in the community, such as emergency shelter, emergency response time, emergency material stockpile points, firefighting materials, household stockpile, emergency plans and drills, emergency science and education activities, and community safety culture construction, which need to be strengthened.

Emergency shelter is a resettlement measure for disaster victims in case of emergencies, and is a project for the benefit of the people to ensure rapid and orderly evacuation and resettlement of people, and minimize casualties and property losses in the event of sudden disasters. Therefore, the community should contact the local government to improve the community emergency shelter construction and the corresponding facilities supporting the emergency shelter system, including (1) the preparation of emergency shelter construction guidelines, the development of multi-hazard comprehensive emergency shelter standards and norms, combined with the characteristics of local disasters and accidents in previous years, as well as the distribution of community population, reasonable planning and construction, can cover the entire community with a certain range of the surrounding emergency shelters, building disaster accident coordination, reasonable layout, and sort management of the emergency shelter system. (2) Improve the supporting facilities of emergency shelters, such as setting emergency shelter signage, and providing water supply, power supply, and communication facilities under emergency conditions. (3) Carry out a census on the construction of emergency shelters, evaluate and determine new emergency shelters, and register them for the record.

Time and speed are crucial in emergency response because complex disaster events are characterized by sudden onset, non-linear amplification, and rapid spread. It is necessary to have fast and flexible emergency response capabilities in response to such disaster events. First of all, it is important to improve the implementation of the emergency response mechanisms, adhering to the principle of “it’s better to take precautions than to lose precautions” according to the worst disaster accident situation, to improve the emergency response plan while establishing the first-time response mechanism. Second, for geological disasters, the meteorological department should strengthen the forecast and early warning of disasters, and other departments should make full use of various channels and ways to release early warning risk aversion information to the community. Finally, the community disaster information team is strengthened, does a good job of disaster hazard investigation, disaster prevention, and avoidance of knowledge propaganda and timely reporting of disaster information during emergencies, and coordinating the relocation, emergency rescue, and rehabilitation work.

Emergency supplies reserve-points, firefighting supplies, and household reserves are all categorized under emergency supplies security. Emergency material security is an important basic work to coordinate the two major issues of development and security and to prevent and resolve major risks. The process is as follows: first of all, select the location of material reserve construction through scientific assessment and decision-making to ensure that the materials can reach the designated location quickly and safely when a disaster occurs. Second, make good reserves of materials, including community reserves of materials and common household reserves of materials, to ensure that resources can meet the temporary response needs of unexpected disasters. Finally, strengthen the establishment of a combination of civilian and combat material reserve mechanisms, the emergency supplies reserve, scheduling process, institutionalization, standardization, division of labor, and responsibility to the person.

Emergency plans and drills, emergency science education activities, and community safety culture construction are all aspects of emergency literacy. To enhance the risk awareness of all people: (1) strengthen the construction of a community safety culture. The influence of culture on people is subtle, invisibly enhancing people’s safety awareness, coordinating people’s relationships, and regulating their behavior, then fulfilling the essential safety of the community. (2) Raise awareness among community residents about the importance of emergency management. We should do a good job in disaster prevention and mitigation publicity, the process of publicity should be close to residents’ lives, and the publicity methods should be innovative, such as holding regular online or offline “safety lectures” and “emergency science lectures”. This will attract residents to accept it voluntarily. (3) Organize regular disaster avoidance drills to ensure that residents know the location and functions of their work and life around the disaster prevention infrastructure. In addition, the results of each drill should be analyzed and organized by the relevant community personnel. The inadequacies of the drilling process should be highlighted in the daily publicity.

## 4. Conclusions

A combination of qualitative and quantitative analysis is proposed for the assessment of the emergency management capacity of resilient communities. Firstly, a model for the evaluation of community emergency management capacity from the perspective of resilience with six primary indicators and 25 secondary indicators is established, and secondly, a community in Xi’an is selected as an example for analysis and research. Due to the uncertainty and coupling effects between various hazards, it is unreasonable not to consider individual perceptions and the integrity assessment of indicators in the actual emergency response capacity building. Therefore, we use the AHP-TOPSIS method to identify the shortcomings in the process of building the emergency management capacity of resilient communities, as a way to improve community resilience building.

It is important to emphasize that the system of emergency management capacity in resilient communities is a complex and dynamic mechanism. It is inappropriate to rely on any single factor to improve community emergency management capacity. An additional improved model or framework is needed here for future research to better assess the correlation between the factors.

## Figures and Tables

**Figure 1 ijerph-19-16201-f001:**
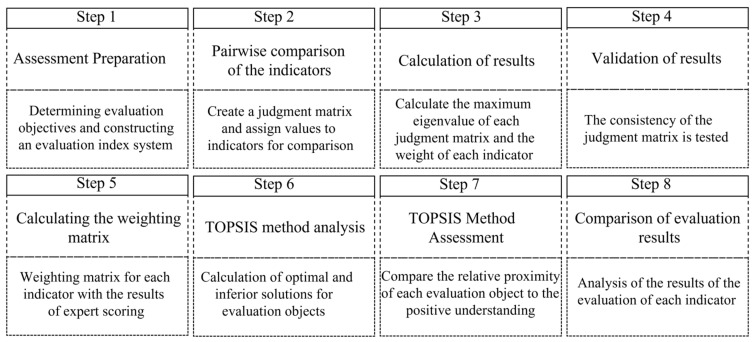
Flow chart of the evaluation process.

**Figure 2 ijerph-19-16201-f002:**
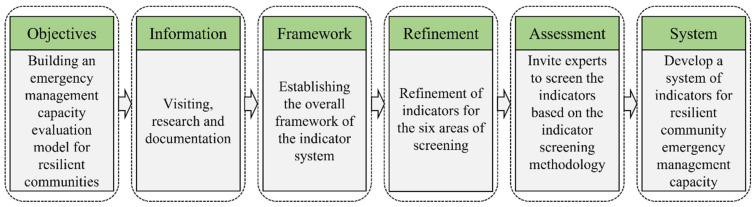
Flowchart for establishing an indicator system.

**Figure 3 ijerph-19-16201-f003:**
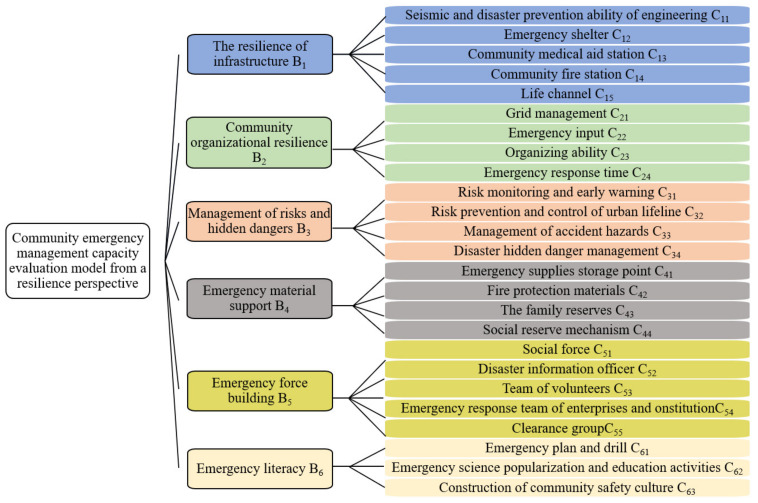
Community emergency management capacity evaluation indicators from a resilience perspective.

**Figure 4 ijerph-19-16201-f004:**
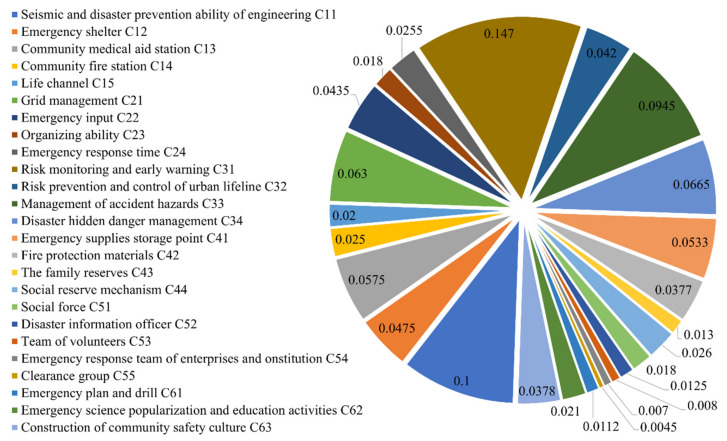
The relative weight of the evaluation index.

**Figure 5 ijerph-19-16201-f005:**
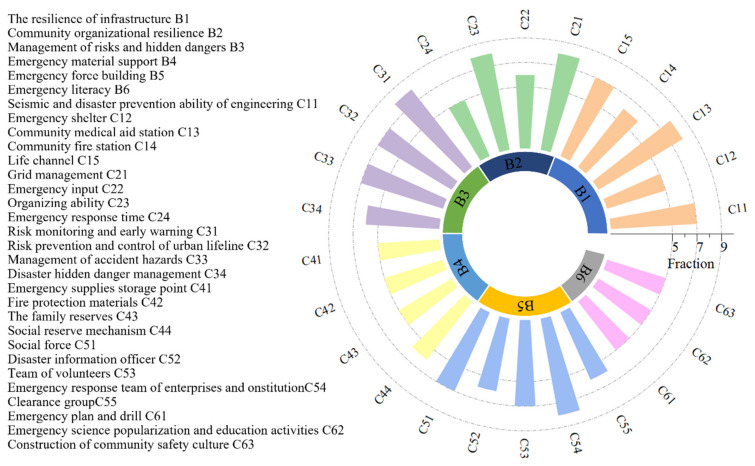
Histogram of community emergency management capacity evaluation results.

**Table 1 ijerph-19-16201-t001:** Judgment matrix scales and their meanings.

Scales	Meanings
1	Indicates that the two factors are of equal importance compared to each other.
3	Indicates that one factor is slightly more important than the other when compared to two factors.
5	Indicates that one factor is significantly more important than the other when compared to two factors.
7	Indicates that one factor is enormously more important than the other when compared to two factors.
9	Indicates that one factor is more important than the other extreme when compared to two factors.
2, 4, 6, and 8	Denotes the median of the above two adjacent judgments.
1/*b_ij_*	Factor *I* is compared with *j* to get judgment *b_ij_*, and *j* is compared with *i* to get judgment *b_ji_*=1/*b_ij._*

**Table 2 ijerph-19-16201-t002:** The unexpected consistency index.

The Matrix Order	3	4	5	6	7
*RI*	0.58	0.89	1.12	1.26	1.36

If *CR* < 0.1, the judgment matrix is considered to have satisfactory consistency; otherwise the judgment matrix needs to be adjusted appropriately until *CR* < 0.1 is satisfied.

**Table 3 ijerph-19-16201-t003:** A-B judgment matrix.

A-B	B_1_	B_2_	B_3_	B_4_	B_5_	B_6_
B_1_	1	3	1/2	2	3	4
B_2_	1/3	1	1/3	3	2	3
B_3_	2	3	1	4	3	4
B_4_	1/2	1/3	1/4	1	5	3
B_5_	1/3	1/2	1/3	1/5	1	1/2
B_6_	1/4	1/3	1/4	1/3	2	1

## Data Availability

The data that support the findings of this study are available from the corresponding author upon reasonable request.

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
