# Peer review of "Study on the Evaluation of Emergency Management Capacity of Resilient Communities by the AHP-TOPSIS Method"

_ijerph, 2022, doi:10.3390/ijerph192316201_

Round 1
Reviewer 1 Report
Review of the paper: ijerph-2064150:
Study on the evaluation of emergency management capacity of resilient communities by the AHP-TOPSIS method
In this paper, the author (s) are interested in emergency management capabilities and the readiness of the related communities. The evaluation phase is performed using two combined technics in the decision-making field. These two technics are the Analytic Hierarchy Process (AHP) and the Technique for Order Preference by Similarity to an Ideal Solution (TOPSIS). The validation of the presented technic AHP-TOPSIS is performed over a real-world case study. The obtained results are satisfactory and provide the decision-makers with a solid background for future decisions in the emergency management area.
Major comments
My First major concern about this paper is the mathematical presentation of the AHP and the TOPSIS methods (lines 113-129; lines 133-162). Indeed, there are several shortcomings that do not make the reading of this paper easy and autonomous. I recommend to the authors the presentation of the related mathematical formulas with clear notations and definitions (especially equations: (1); (3); (7); and (8)). In addition, there is no single reference to the AHP and TOPSIS methods. Therefore, it is highly recommended including some recent references about these two methods.
The second major concern is about the scientific contributions of this paper. In this context, the authors should clearly state the contributions of this work compared to the existing literature.
Minor comments
1. Line 12: (TOPSIS) should be defined in line 12 and not in line 90. The same comment is valid for the AHP method.
2. Lines 38-87: The paragraph is too long. Try to subdivide it into more paragraphs.
3. Last row of Table 1: replace “and j is compared with I to” by “and j is compared with i to”.
4. Line 114: define the symbols λmax and wi
5. Line 117: define w*ij in equation (1).
6. Line 121: define Aw in equation (3).
7. Line 150: replace “Where, I is a..” by “Where, i is a..”.
8. Line 300 (conclusion): Please include some indications about the future research work.
Reviewer 2 Report
The authors employed AHP-TOPSIS method for the evaluation of emergency management capacity. It is a very small article. I found nothing new in the paper. My major comments are given below:
1. The study is mainly based on a community emergency management capacity evaluation index system with six primary indicators and 25 secondary indicators. The process used to develop the index system is discussed in 2/3 sentences in lines 167-172. But it is not a simple issue. The evaluation index is the core of capacity management. Generally, experts, community, policymakers, operational management staff, etc. are involved through various processes, including focus group discussions, workshops, questionnaire surveys, etc. The initial selection of factors and finally coming to a reduced number of factors is a tedious process. The author completely avoided the process of deciding the evaluation index system. If it is an existing evaluation system, there is no problem with using it. But if you developed it, you must have to discuss it clearly.
2. What is the novelty of the study? It seems from the last paragraph of the introduction that the authors want to tell the combination of the AHP and TOPSIS as the study's novelty. But the whole introduction author has not reviewed any literature related to similar studies where the combination of multi-criteria decision-making (MCDA) methods are used to reduce subjectivity. The introduction is very monotonic. Just informed who found which factors affect community resilience. I suggest rewriting the Introduction section. Compact the second paragraph. Add a new paragraph to describe previous emergency management capacity evaluation studies using MCDA. Try to emphasize how your study is different from the others.
3. It is mentioned that a total of nine experts scored the importance judgment of each index. Can you please elaborate a bit in detail? How are the experts selected? What is their expertise? What process is used in scoring the index?
4. It is mentioned at the end of the introduction: "...effectively reduce the influence of subjective factors". But it is not clear after reading the paper how it reduced the subjectivity.
5. Conclusion of the article can not be considered a conclusion. They just repeated what they intended to do.
Reviewer 3 Report
Please find the detailed comments attached.

Round 2
Reviewer 1 Report
All my comments are addressed and I recommend the acceptance of the paper.

Author Response
Thank you for your kind work.
Reviewer 2 Report
-
Author Response
Thank you for your kind work.
Reviewer 3 Report
Review report of the revised version of article ijerph-2064150
The paper titled “Study on the evaluation of emergency management capacity of resilient communities by the AHP-TOPSIS method” is reconsidered after major changes. The paper improved significantly in brief, and the rest of the comments are addressed.
The second version of the manuscript has some minor problems; before publication, please deal with the followings:
(1) In Section 3.4 there are no linkages between the results of the research and the results of others’ work. If possible, link the exact results or some parts of the results or the methodological novelty of the manuscript to other articles and highlight the relative importance of the work in the light of other researches.
(2) The newly added references [54] and [58] are the same. Please solve this problem (if it has relevant issues in the text, please address and solve them in the text as well.)
(3) Please change the authors in reference [51] to “Bognár, F., Benedek, P.”
Overall evaluation:
The work has significant potential and is worth publishing in IJERPH after minor changes based on the previously discussed comments.
2022.11.30.
